# The Effects of Different Doses of Sildenafil on Coronary Blood Flow and Oxidative Stress in Isolated Rat Hearts

**DOI:** 10.3390/ph16010118

**Published:** 2023-01-13

**Authors:** Nada Banjac, Velibor Vasović, Nebojša Stilinović, Ana Tomas, Lucija Vasović, Nikola Martić, Dušan Prodanović, Vladimir Jakovljević

**Affiliations:** 1Medical Faculty, University of Banja Luka, 78000 Republika Srpska, Bosnia and Herzegovina; nada.banjac@teol.net; 2Department of Pharmacology, Toxicology and Clinical Pharmacology, Faculty of Medicine, University of Novi Sad, 21000 Novi Sad, Serbia; velibor.vasovic@mf.uns.ac.rs (V.V.); nebojsa.stilinovic@mf.uns.ac.rs (N.S.); ana.tomas@mf.uns.ac.rs (A.T.); nikola.martic@mf.uns.ac.rs (N.M.); 3Faculty of Medicine, University of Novi Sad, 21000 Novi Sad, Serbia; 902005d22@mf.uns.ac.rs; 4Department of Physiology, Faculty of Medical Sciences, University of Kragujevac, 34000 Kragujevac, Serbia; drvladakgbg@yahoo.com

**Keywords:** sildenafil, phosphodiesterase, isolated rat heart, oxidative stress, coronary flow

## Abstract

The dose-response relationship of sildenafil effects on cardiac function is not completely elucidated. The aim of this study was to assess the effects of different doses of sildenafil on coronary flow and oxidative stress in isolated rat hearts. Coronary flow and markers of oxidative stress, including nitrite outflow, and superoxide anion production in coronary effluent, were determined for isolated rat hearts. The experiments were performed during control conditions and in the presence of sildenafil (10, 20, 50, 200 nM) alone or with Nω-nitro-L-arginine monomethyl ester (L-NAME) (30 μM). Sildenafil was shown to result in a significant increase in coronary flow at lower coronary perfusion pressure (CPP) values at all administered doses, whereas, with an increase in CPP, a reduction in coronary flow was observed. An increase in nitric oxide (NO) was most pronounced in the group treated with the lowest dose of sildenafil at the highest CPP value. After the inhibition of the NO-cyclic guanosine monophosphate (cGMP) signaling (NOS) system by L-NAME, only a dose of 200 nM sildenafil was high enough to overcome the inhibition and to boost release of O_2_^−^. That effect was CPP-dependent, with statistical significance reached at 80, 100 and 120 mmHg. Our findings indicate that sildenafil causes changes in heart vasculature in a dose-dependent manner, with a shift from a vasodilatation effect to vasoconstriction with a pressure increase. The highest dose administered is capable of producing superoxide anion radicals in terms of NOS system inhibition.

## 1. Introduction

Sildenafil, the first drug developed for the treatment of erectile dysfunction, is a highly selective inhibitor of phosphodiesterase (PDE) type 5 [1]. By inhibiting phosphodiesterase, it increases the intracellular concentrations of cyclic guanosine monophosphate (cGMP), causing an amplification of the endogenous nitric oxide (NO)-cGMP signaling (eNOS) pathway, thus enhancing the NO-mediated blood vessel relaxation [2]. After accumulation, cGMP interacts with several targets, including the cGMP-dependent protein kinase (PKG). Emerging evidence indicates that drugs that increase cGMP concentrations may have important applications in the treatment of a variety of cardiovascular disorders, such as atherosclerosis and hypertension [3].

Recent experimental studies show that sildenafil exhibits a preconditioning-like cardioprotective effect against ischemia/reperfusion injury in the intact heart. This is not surprising, as the cGMP/PKG pathway has been shown, in many reports, to be involved in the protective signaling of heart preconditioning [4]. The study conducted on humans that tested PDE5 inhibitor, tadalafil, and effects on endothelial function, showed significant differences in the flow-mediated dilatation and nitrite levels, compared to control [5]. Results suggest a link between sildenafil-induced delayed protection and the activation of a NOS-dependent signaling cascade. Sildenafil-induced protection against necrosis and apoptosis was abolished in myocytes derived from inducible nitric oxide synthase (iNOS), but not from eNOS gene knockout mice [6]. Sildenafil produces early and late preconditioning in rabbits in vivo, causes delayed preconditioning in mice, and this effect is primarily mediated by iNOS [7,8]. In rat myocardium, sildenafil decreases myocardial infarct size, enhances functional recovery, and increases cGMP content [9]. Sildenafil-induced infarct-limiting protection was also reported in dogs [10]. These protective properties may be related to its cGMP elevating and cAMP suppressing effects [11,12], while the end-effectors for sildenafil in the ischemic heart include the mitochondrial and sarcolemmal K(ATP) channels [13]. In the model of renal stenosis, the beneficial effects of sildenafil were also reported to be related to reduction oxidative stress and subsequent improvement of viability and a decrease of DNA [14]. Oxidative stress has also been implicated as an important factor in cardiovascular diseases, and the effects of sildenafil in cardiovascular diseases are pleiotropic and may also involve changes of the pro-/antioxidant balance, lipid peroxidation, and autonomic control [14]. The action of sildenafil is therefore not solely limited to mechanistic inhibition of PDE5-PKG pathway, but also includes alteration of the inflammatory and redox status [15].

In contrast, a number of studies failed to establish a clear relationship between sildenafil use and cardiac effects in a model of ischemia/reperfusion injury. This discrepancy may be attributed to differences in the dose of the drug used [13]. A similar study performed on isolated rat hearts showed that low concentrations of sildenafil (20–50 nM) improve reperfusion function, while higher concentrations (200 nM) worsen it [6]. The dose-response relationship of sildenafil effects on cardiac function and various stress markers is not completely elucidated. For this purpose, the aim of our study was to evaluate the effects of different sildenafil doses on coronary flow, nitrite levels, and oxidative stress markers in the isolated rat heart, with possible impact on the endothelial L-arginine/NO system.

## 2. Results

Impact of different sildenafil doses alone on cardiac blood flow is presented in Figure 1.

There were statistically significant differences between control measurements and measurements after perfusion with sildenafil. Control measurements showed that an increase in CPP is followed by an increase in cardiac blood flow, while a rise in CPP led to a decrease in coronary flow (CF) after perfusion with sildenafil. At CPP less than 80 mmHg, the CF values were significantly higher in the sildenafil-treated groups, whereas at perfusion pressures greater than 80 mmHg, the effect was completely the opposite, with considerably lower CF values in the same groups. Furthermore, only in the sildenafil 20 nM group was CF was notably higher than the control at CPP of 80 mmHg. The addition of L-NAME reversed these sildenafil-induced changes at sildenafil doses of 10 nM and 200 nM, with lower CF observed at all pressures without reaching statistical significance. Conversely, addition of L-NAME to sildenafil at doses of 20 nM and 50 nM showed the same pattern as for sildenafil alone, with CF significantly higher at CPP less than 80 mmHg and CF significantly lower at CPP greater than 80 mmHg (Figure 2).

The effect of sildenafil alone and following L-NAME pre-treatment on the nitrite outflow can be seen in Figure 3 and Figure 4.

Statistical significance was reached in sildenafil 10 nM group at the highest recorded pressure (13.55 ± 10.46 vs. 34.45 ± 17.91) and sildenafil 50 nM group at the lowest pressure (1.27 ± 1.31 vs. 6.59 ± 3.75). Only the group with sildenafil 20 nM + L-NAME had significantly higher level of nitrite than control.

Administration of sildenafil in doses of 10 nM and 200 nM released lower concentrations of O_2_^−^ at all pressures compared to control, but without statistical significance (Figure 5 and Figure 6).

The perfusion with a combination of L-NAME and sildenafil at doses of 10 nM, 20 nM and 50 nM generated lower O_2_^−^ values in comparison to control at all pressures. The opposite could be seen for sildenafil 200 nM + L-NAME group, where statistically significant higher O_2_^−^ release was reached at CPP of 80 mmHg (13.23 ± 8.09 vs. 33.19 ± 8.30) and 120 mmHg (24.99 ± 11.41 vs. 50.41 ± 10.36).

## 3. Discussion

By observing changes in coronary flow in a function of CPP, sildenafil was shown to result in a significant increase in coronary flow at lower CPP values at all administered doses, whereas, with an increase in CPP, a significant reduction in coronary flow was observed. In other words, sildenafil decreases coronary flow, likely by increasing myocardial cGMP level in a dose- and pressure-dependent manner. This result is very interesting and is inconsistent with some of our previous studies. For example, in an experiment with vardenafil, we found that it did not influence coronary circulation at doses from 10 nM to 200 nM, which are similar doses to those used here for sildenafil [16]. Furthermore, this can be a significant finding, given the pronounced vasodilator effect under lower pressures and the paradoxical vasoconstrictor effect at higher CPP conditions. In addition, some of the studies demonstrated that sildenafil possesses a cardioprotective action by not only affecting coronary circulation but changing ischemic post-conditioning. One of the first experiments dealing with the effect of sildenafil on an isolated heart showed reduction in myocardial infarct size for all doses administered [7]. Various authors have tried to confirm these results, but with partial success. Reffelman and Kloner (2003) concluded that sildenafil seems to be neutral and safe in terms of myocardial function; however, according to the results of our study, that can be true for lower perfusion pressures [17].

A review of the literature showed that the measured values for the potency and selectivity of PDE-5 inhibitors vary, with the IC50 values on PDE5 for sildenafil ranging from 3.5 nM to 8.5 nM [18]. We have selected our concentration ranges based on previous studies and in consideration of the IC50 values from the literature [11], with the minimum tested concentration expected to induce PDE5 inhibition and provide a measurable effect in tested parameters in our study design. Regarding the obtained differences in the results of our study for two close concentrations, we must again make clear that each heart was tested only for a single dose of sildenafil; therefore, for these two close concentrations, differences might have been the consequence of different basal levels of cGMP or activity of guanylyl cyclase and other steps in the signaling cascade. However, with higher tested concentrations (50–200 nM), differences between control and experimental conditions were more consistent. In the interpretation of findings, we have now included information that observed effects suggest that effects of sildenafil may involve not only PDE5 inhibition, but also the inhibition of other PDE types. It was already reported in the literature that the increase in sildenafil dose and concentration decreases its specificity for only PDE5 inhibition, and other PDE subtypes could be inhibited [18]. The highest concentration of sildenafil that we used in our experiment (200 nM) showed negative effects on coronary flow, which is similar to the results of the other studies, where a 50 nM concentration improved reperfusion function and reduced infarct size, while the concentration of 200 nM led to exacerbation of ischemic/reperfusion injury [11].

Using the same model and same dose range as ours, Das et al. (2005) noticed that positive effect depends on sildenafil’s concentration, with vasodilatation at lower concentrations (10, 20 and 50 nM) and vasoconstriction at the highest concentration applied (200 nM) [6]. Moreover, they measured myocardial cGMP and discovered that lower sildenafil doses caused cGMP rise and hypothesized that the end-effect of sildenafil is the opening of the sarcolemmal and mitochondrial potassium channel. Salloum et al. (2003) went further and proved that sildenafil induces cardioprotective preconditioning, mainly through inducible NO synthase-dependent pathway [19]. This was inhibited with by nitric-oxide synthase inhibitor, L-NAME. Thus, those two facts could explain why the inhibition of the NO system by L-NAME completely abolished previously presented effects for the lowest and the highest sildenafil dose and stayed practically the same for mid-doses (20 and 50 nM) in our experiment [8]. Recent studies again documented sildenafil’s safety and underlined that by indicating a reduction in cardiovascular mortality and an improvement of symptoms in heart failure in PDE5 users [20].

NO is the molecule that can trigger the cardioprotective cGMP-PKG pathway. NO first activates soluble guanylate cyclase (sGC), which catalyzes synthesis of cGMP. Elevated intracellular level of cGMP mostly targets and activates cGMP–dependent protein kinase (PKG). PKG then lower the blood pressure through vasodilatation [21]. As the ischemia effects on the content of myocardial cGMP, it is shown, in the isolated heart model, that cardiomyocytes have the ability to increase cGMP levels during the first 10–25 min of ischemia, with an acute decrease after that period [22]. Although a number of studies have tried to stimulate the cGMP synthesis, preservation of cGMP by blockade of PDE5 is demonstrated to limit the infarct size in different studies [23]. Despite the fact results from the study conducted by Elrod et al. [24] showed no elevation of cGMP levels after sildenafil administration, they hypothesize that subcellular localization of various cGMP pools (soluble cGMP and particulate cGMP) could have a significant place in the cGMP/PKG pathway of cardioprotection [24]. By observing changes in NO release in a function of the increase in coronary flow, our results show that sildenafil led to an increase in NO release in all doses except the highest. This increase was most pronounced in the group treated with the lowest dose of sildenafil at the highest CPP value. After the inhibition of the NOS system by L-NAME, these effects were reduced at all CPP values. Interestingly, nitrite increment were retained in the group treated with 20 nM sildenafil, despite the inhibition of the NOS system by L-NAME. This may be the consequence of the sildenafil-mediated cardioprotection, independent of the iNOS, eNOS and the cGMP pathway, as tested in the study from United States [24]. In that study, they used iNOS and eNOS knockout mice, in which they induced an infarct and treated them with sildenafil afterwards [24]. Their results showed a reduction in infarct per area at risk (Inf/AAR) in mice, despite the lack of iNOS or eNOS [24], suggesting that sildenafil cardioprotective mechanisms are not only iNOS and eNOS related. Thus, the inhibition of nitric oxide synthase system by L-NAME was overcame by those other mechanisms of action of sildenafil, which resulted in an increase of NO concentration in the experimental group compared to the control group. Considering this, the reason why sildenafil had significant effects on the NO concentration was only observed in the 20 nM group, and not in the others, could be explained by L-NAME wide, nonspecific effects, besides the nonspecific inhibition of NOS [25]. Potential direct effects on cardiac tissue, as hypothesized by some authors [25], could be one of those properties of L-NAME that interfere with the cardioprotective effects of sildenafil.

Sildenafil’s pleiotropic action ranges from clinically proved pulmonary artery vasodilation to various antioxidative effects, which lead us to hemodynamic of released nitrites and superoxide anion radical. This could be a link to another possible mechanism of sildenafil’s action [20,26,27]. Despite some other control mechanisms, according to available data, the predominant mechanism of NO degradation in vivo is the conversion to nitrates (NO^3−^) in reversible reaction with oxyhaemoglobin. Another important mechanism of NO reduction is reaction with superoxide anion radical (O^2−^). Endothelial L-arginine/NO system, besides NO, continuously produces physiological amounts of O^2–^. Overproduction of O^2–^ is usually accompanied with rapid NO degradation to form peroxynitrite (ONOO^–^) in a reaction which is 30–50 times faster than the action of superoxide dismutase (SOD), one of the most powerful antioxidant enzymes which scavenges superoxide anion radical. That newly formed peroxynitrite anion is an oxidizing agent capable of destroying cell membranes [14,28,29]. With regard to changes in O^2−^ released in a function of the coronary flow increase, our results show that sildenafil did not affect significantly the release of O^2−^, with the only exception being doses of 20 and 50 nM at the lowest CPP. After the inhibition of the NOS system by L-NAME, it became evident that only a dose of 200 nM sildenafil was high enough to overcome the inhibition and to boost the release of O^2−^. That effect was CPP-dependent, with statistical significance reached at 80, 100, and 120 mmHg. The same dose of sildenafil did not change CF and nitrite concentration at statistically significant levels after the NOS suppression; however, there were lower values for CF and nitrite concentration at all CPPs. Thus, we hypothesize that the highest sildenafil dose can produce superoxide anion radicals by NOS in a dependent mechanism, which may negatively influence coronary flow. The rationale behind this could be that another enzyme dealing with L-arginine is involved, but our study is lacking teomics to prove this. That enzyme could be arginase, which inhibits endothelial and inducible NOS synthase activity and consequentially diminishes NO production. On the other hand, NO scavengers, superoxide, and peroxynitrite are unopposed [30]. Many authors are consentient that the main reason for vascular dysfunction is the over-production of superoxide radicals, which in turn reacts with NO to form reactive nitrogen species, effectively reducing NO bioavailability [20,26,31,32,33]. It must be noted that, besides the results of our group, no comparable data use the same biochemical assays for measuring NO activity and oxidative stress markers in the isolated rat heart model. However, in those experiments, we used other substances capable of changing NO and O^2−^ levels, like nimodipine and DL-homocysteine thiolactone, not sildenafil [29,34,35].

Previous studies have reported on the rate reduction effect of sildenafil, partially attributed to the NO synthase-mediated signaling pathways, shown to play an important role in pulmonary vein arrhythmogenesis [36]. Sildenafil was also shown to have a negative chronotropic effect on mice sino-atrial nodes [37]. This implies that sildenafil may potentially protect from atrial fibrillation. On the contrary, some literature suggests that sildenafil alone, as well as the combination of sildenafil and L-NAME, does not affect heart rate in rats [38,39]. Furthermore, higher concentrations of sildenafil (0.2 µg/mL), with a nitric oxide (NO) donor, increased ventricular tachycardia or fibrillation, implying a potential risk of atrial arrhythmogenesis [40]. Additionally, studies conducted on dogs and humans found that sildenafil does not affect cardiac contractility (inotropism) [41], despite the fact that, in addition to direct effects on cardiomyocytes, inhibitors of different PDE subtypes regulate vascular activity, as well as cardiac contractility. However, sildenafil was found to increase the pulmonary vein diastolic tension, but decrease contractility, which was attributed to the electrophysiological effects of sildenafil via mechanoelectrical feedback [36]. Additionally, PDE5A inhibition by sildenafil was found to blunt systolic responses to beta-adrenergic stimulation in a clinical study, supporting the role of the PDE5A activity in modifying stimulated cardiac function [42].

Despite so much controversial data about sildenafil’s effect on heart and blood vessels it still represents an interesting drug to study, which is proven by many recent preclinical and clinical studies [32,33,43,44]. Taking into account that coronary vasculature fills during the diastole (80 mm Hg) [45], results observed at CPP 60–100 mm Hg should have the greatest implications for future clinical studies, with 10 nM and 20 nM concentrations being used as a starting dosage of sildenafil, considering that most of the statistically significant results from our study were observed with those doses at 60–100 mm Hg of CPP.

## 4. Materials and Methods

### 4.1. Chemicals and Reagents

Sildenafil, L-NAME of analytical grade, and all reagents for spectrophotometric assays kit were purchased from Sigma-Aldrich Chemie GmbH, Schnelldorf, Germany. Reagent kit for Krebs-Hensenleit perfusion solution was purchased from Merck, Vienna, Austria.

### 4.2. Experimental Procedure

The hearts of male Wistar albino rats (*n* = 48, 6 in each experimental group, weighing 180–200 g) were excised and perfused according to the modified Langendorff technique at constant pressure conditions (Experimetria Ltd., Budapest, Hungary), as described previously [28,34,46]. The animals were purchased from the Military-Technical Institute in Belgrade. The animals were maintained under controlled room temperature (23 ± 1 °C) and a light and dark (12:12 h) cycle, with access to food and water *ad libitum*. Animal care and all experimental procedures were carried out in accordance with the ethical principles of working with laboratory animals given in the Guide for the Care and Use of Laboratory Animals. The Ethics Committee of Medical Faculty, University of Kragujevac for Experimental Animals, has given approval for the study (Approval No. 01-6490; 4 November 2008).

On the day of the experiment, the animals were premedicated with heparin, put under ether anesthesia and sacrificed by cervical dislocation. Immediately after, thoracotomy was performed, and rapid cardiac arrest was induced by superfusion with ice-cold isotonic saline. The hearts were excised and placed on a *Langendorff* apparatus; aortas were cannulated and retrogradely perfused using complex Krebs-Henseleit perfusate at the constant pressure (CPP) of 60 cm H_2_O. The composition of the nonrecirculating Krebs-Henseleit perfusate was as follows (mmol/L): NaCl 118, KCl 4.7, CaCl_2_×2H_2_O 2.5, MgSO_4_×7H_2_O 1.7, NaHCO_3_ 25, KH_2_PO_4_ 1.2, glucose 11, pyruvate 2, equilibrated with 95% O^2^ plus 5% CO^2^ and warmed to 37 °C (pH 7.4). Sildenafil solution was made by dissolving sildenafil in dimethyl sulfoxide (DMSO) and diluting with perfusion buffer to the final concentration of DMSO of 0.001%. For control buffer solution, the same volume of DMSO was added. Coronary flow (CF) was measured using the flowmetric method. Constant left ventricular draining was performed through resected mitral valve. Namely, by using the sensor (BS4 73-0184) placed in the left ventricle, we continuously registered parameters of myocardial function. After the equilibration period, coronary perfusion pressure was lowered to 40, 30 and 20 cm H_2_O and then gradually increased to 80, 100, 110 and 120 cm H_2_O, in order to establish coronary autoregulation. When the flow was considered stable at each of the tested perfusion pressures, samples of the coronary effluent were collected. At the end of the varying pressures [47] (basic protocol), isolated hearts were perfused with:(1)10 nM sildenafil.(2)20 nM sildenafil.(3)50 nM sildenafil.(4)200 nM sildenafil.(5)Five-minute perfusion with 30 μM L-NAME, followed by perfusion with a combination of 10 nM of sildenafil and 30 μM L-NAME.(6)Five-minute perfusion with 30 μM L-NAME, followed by perfusion with a combination of 20 nM sildenafil and 30 μM L-NAME.(7)Five-minute perfusion with 30 μM L-NAME, followed by perfusion with a combination of 50 nM sildenafil and 30 μM L-NAME.(8)Five-minute perfusion with 30 μM L-NAME, followed by perfusion with a combination of 200 nM sildenafil and 30 μM L-NAME.

The measurements were taken at baseline, prior to administration of any medication at different pressures (40, 60, 80, 100 and 120 mm Hg). After the highest CPP (120 mm Hg) of control testing, the pressure was recovered to the base point (40 mm Hg), followed by a stabilization period of 5 min, after which constant administration of phosphodiesterase 5 inhibitor sildenafil solution started. After another 5 min, when steady CF value has been reached, CF value was recorded. Then, CPP was raised for 20 mm Hg; another 5 min passed before measurements; and so on, until the CPP of 120 mm Hg, when the experiment for one isolated rat heart would end. Every rat heart was tested with only one sildenafil or sildenafil + L-NAME dose. In the second series of experiments, basic protocol described above was followed by perfusion with Nω-nitro-L-arginine monomethyl ester (L-NAME), as an inhibitor of NO synthase from the start of the experiment. After the 5 min of constant L-NAME perfusion, sildenafil was added, and continuation of experiment was the same as with sildenafil alone, with constant measurements of myocardial parameters (Figure 7).

### 4.3. Nitrite and Superoxide Anion Radical (O^2−^) Determination

Samples of the coronary venous effluent were collected following the stabilization of flow at each of the gradually increased perfusion pressures. Parameters of oxidative stress were determined through spectrophotometric methods. Nitrite levels served as an index of nitrite oxide production and were determined by the spectrophotometric method using the Griess’s reagent [7]. Briefly, 0.5 mL of perfusate was precipitated with 200 µL of 30% sulfosalicylic acid, vortexed for 30 min and centrifuged at 3000× *g*. Equal volumes of the supernatant and Griess’s reagent, containing 1% sulphanilamide in 5% phosphoric acid/0.1% naphthalene ethylenediamine-dihydrochloride, was added and incubated for 10 min in the dark and read at 543 nm. Sodium nitrite served as a standard for calculation of nitrite levels. The levels of the superoxide anion radical (O^2−^) were measured through Nitro Blue Tetrazolium (NBT) reaction in TRIS-buffer. The assay mixture contains: 50 mM TRIS-HCl buffer (pH = 8.6), 0.1 mM EDTA, 0.1 mg/ml gelatin, and 0.1 mM NBT. The reaction was performed by mixing 50 mcl of venous coronary effluent and 950 mcL of assay mixture and reading absorbance at 550 nm. Krebs-Henseleit solution served as a blank probe.

### 4.4. Statistical Analysis

Statistical analysis was performed using IBM SPSS statistical software, version 19.0 (IBM Corp., Armonk, NY, USA). Data were reported as the mean ± standard deviation (SD). Data were assessed for normality using the Kolmogorov–Smirnov test. A paired-samples t-test or Wilcoxon signed-rank test as a non-parametric equivalent was used to compare groups. Furthermore, a mixed between-within subjects analysis of variance was conducted to estimate the effect of different pressures (40, 60, 80, 100, 120 mm Hg) and treatment (control and experimental group). Tukey’s HSD post hoc were performed, and a difference between groups was considered statistically significant at a *p* value less than 0.05 (*p* < 0.05).

## 5. Conclusions

In summary, our findings suggest that sildenafil caused changes in heart vasculature in a dose-dependent manner and shifted its vasodilatation effect to vasoconstriction with a pressure increase. Furthermore, the highest dose administered here (200 nM) is capable of producing superoxide anion radicals in terms of NOS system inhibition. In order to elucidate these results further, proteomics assays, like immunohistochemical, RT-PCR or Western blot, are necessary in future experiments.

## Figures and Tables

**Figure 1 pharmaceuticals-16-00118-f001:**
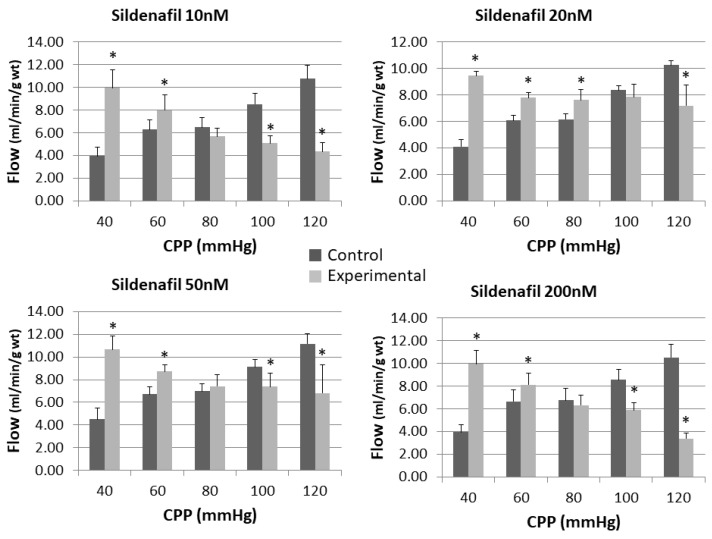
Influence of different sildenafil doses on the coronary flow of the isolated rat heart. * *p* < 0.05, Mixed between-within subject ANOVA; Tukey’s HSD.

**Figure 2 pharmaceuticals-16-00118-f002:**
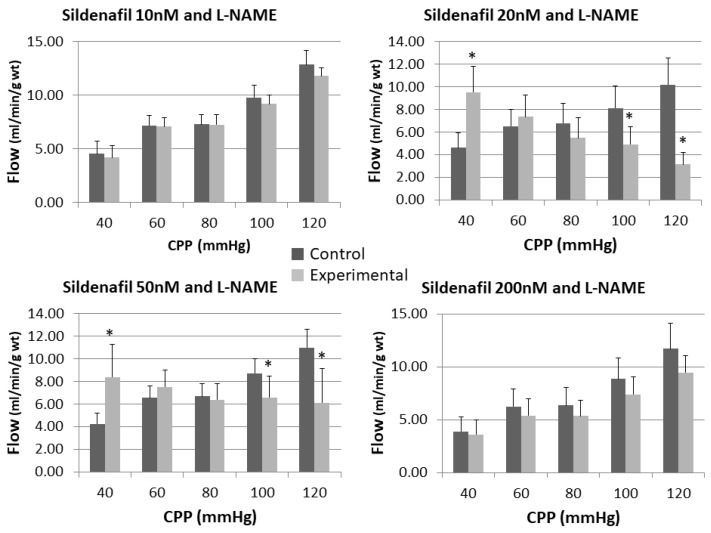
Influence of different sildenafil doses + L-NAME on the coronary flow of the isolated rat heart. * *p* < 0.05, Mixed between-within subject ANOVA; Tukey’s HSD.

**Figure 3 pharmaceuticals-16-00118-f003:**
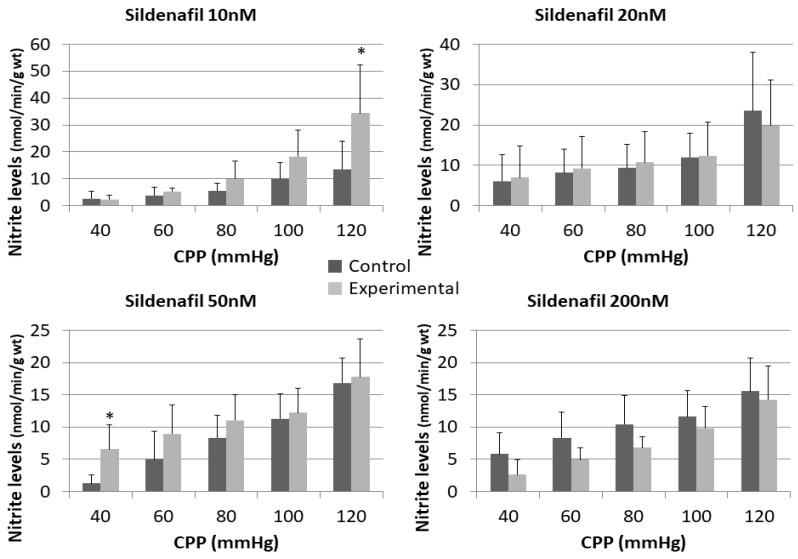
Influence of different sildenafil doses on the dynamics of released nitrite in the isolated rat heart. * *p* < 0.05, Mixed between-within subject ANOVA; Tukey’s HSD.

**Figure 4 pharmaceuticals-16-00118-f004:**
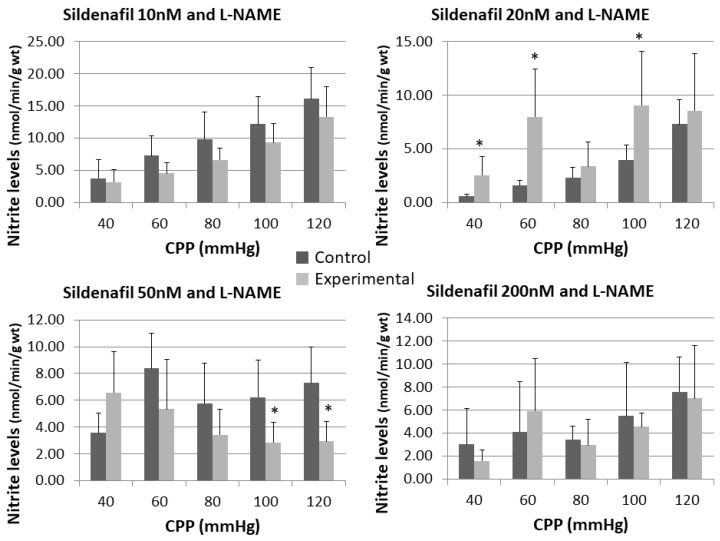
Influence of different sildenafil doses + L-NAME on the dynamics of released nitrite in the isolated rat heart. * *p* < 0.05, Mixed between-within subject ANOVA; Tukey’s HSD.

**Figure 5 pharmaceuticals-16-00118-f005:**
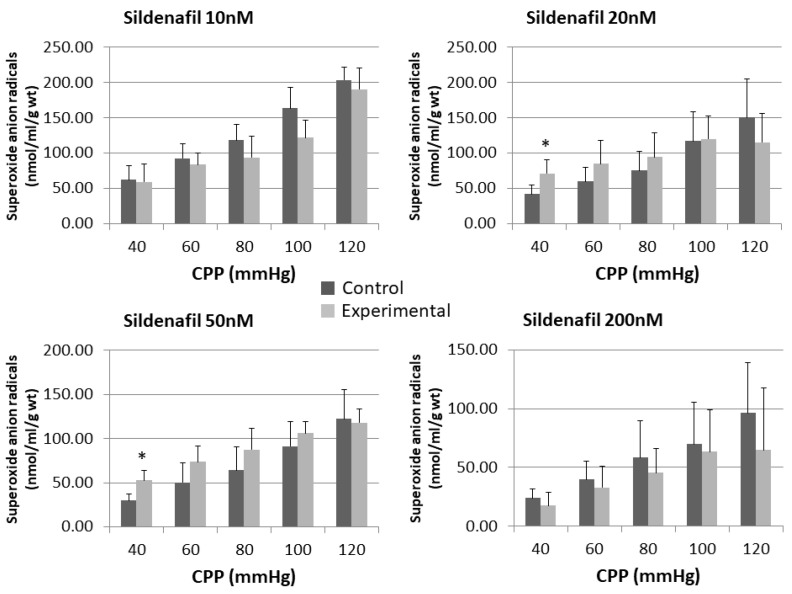
Influence of different sildenafil doses on the dynamics of released superoxide anion radical (O^2−^) in the isolated rat heart. * *p* < 0.05, Mixed between-within subject ANOVA; Tukey’s HSD.

**Figure 6 pharmaceuticals-16-00118-f006:**
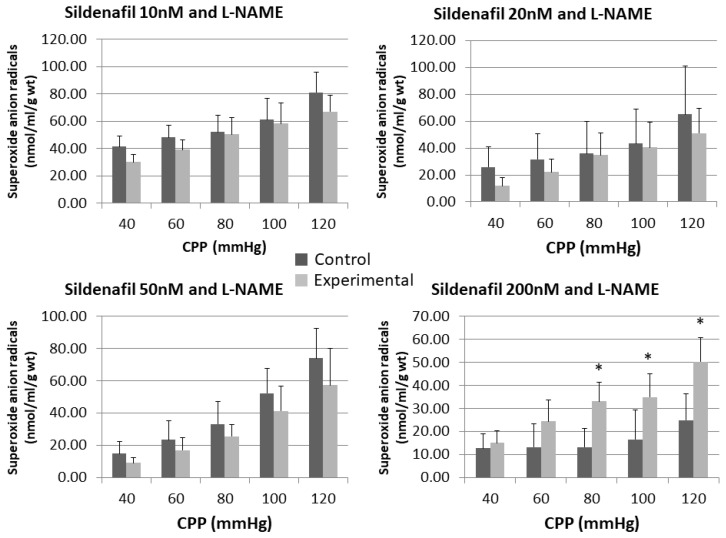
Influence of different sildenafil doses + L-NAME on the dynamics of released superoxide anion radical (O^2−^) in the isolated rat heart. * *p* < 0.05, Mixed between-within subject ANOVA; Tukey’s HSD.

**Figure 7 pharmaceuticals-16-00118-f007:**
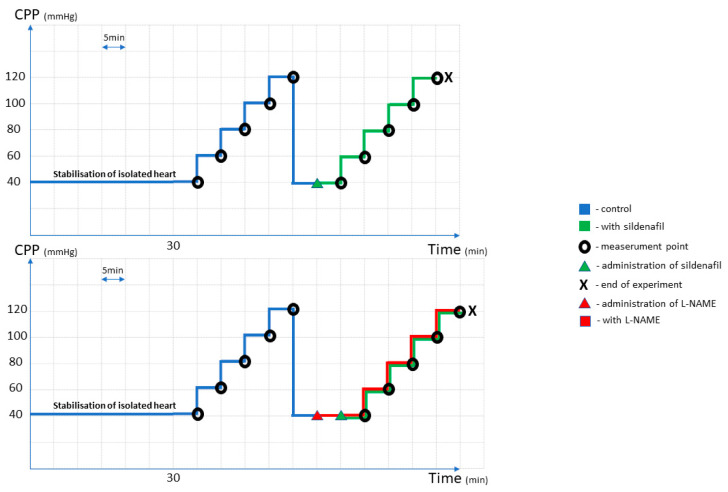
Control and experimental protocol and drug administration scheme, once the heart was isolated from the rat.

## Data Availability

Data is contained within the article.

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
