# Peer review of "The Effects of Different Doses of Sildenafil on Coronary Blood Flow and Oxidative Stress in Isolated Rat Hearts"

_pharmaceuticals, 2023, doi:10.3390/ph16010118_

Round 1

Reviewer 1 Report

This review article lacks novelty. Below are some comments for the authors to check.

1.      Abstract: CPP and NO must write the full name when they appear for the first time.

2.      Page 2 line 75, change x ± SD to mean ± SD.  Line 77, please explain how to compare.

3.      MATERIALS AND METHODS:Page 7 line 238, provide ethics committee approval number and the name of the institution for animal and human studies. Line 245, change H20 to H2O. Line 258, change 10Nm to 10 nM.

4.      Page 8 line 305, delete “All values are expressed as the mean ± SD.”

5.      In the References section, the first word should be capitalized, and the other words should be lowercase, such references 3, 7, 8, 19, and 25. The authors need to update references.

6.      Finally, many experiments are to be done in the future as described by the authors at page 8 lines 312-313 "Proteomics assays like immunohistochemical, RT-PCR or Western blot are necessary in the future experiments". Why not present the results of these experiments int this manuscript? Therefore, this manuscript data too few and less novelty.

Author Response

Response:

Firstly, thank you for taking time to review our manuscript. We appreciate your feed-back as well as the comments that helped us to significantly improve the clarity and quality of our paper. We have thoroughly revised and rewritten our paper in an effort to address each of your suggestions. Please find bellow the point-by-point the changes we have made to the manuscript.

Comment 1:

Abstract: CPP and NO must write the full name when they appear for the first time.

Response:

All abbreviations have been defined at their first appearance in the text, as suggested.

Comment 2:

Page 2 line 75, change x ± SD to mean ± SD. Line 77, please explain how to compare.

Response:

We changed every “x ± SD” to “mean ± SD” wherever it appeared in our manuscript. We have used a mixed between-within subject ANOVA. In the other words, we did comparison between every CPP value (40, 60, 80, 100, 120 mm Hg) as well as between control and experimental group. Statistical significances noted in graphics represent the post-hoc analysis (Tukey’s HSD) between control and experimental group, once the ANOVA was confirmed as statistically significant. We have highlighted it in the legend section of every graph.

Comment 3:

MATERIALS AND METHODS: Page 7, line 238, provide ethics committee approval number and the name of the institution for animal and human studies. Line 245, change H20 to H2O. Line 258, change 10 Nm to 10 nM.

Response:

The ethics committee approval number and the name of the institution for animal and human studies have been stated in the Materials and methods, Experimental procedure section, as you suggested.

We also corrected mistakes that you noticed, from lines 245 and 258.

Comment 4:
Page 8 line 305, delete “ All values are expressed as the mean ± SD.”

Response:

We have deleted the specified sentence.

Comment 5:

In the References section, the first word should be capitalized, and the other words should be lowercase, such references 3, 7, 8, 19 and 25. The authors need to update references.

Response:

The listed references are now presented according to the instructions you suggested. We also updated references.

Comment 6:

Finally, many experiments are to be done in the future as described by the authors at page 8 lines 312-313 “Proteomics assays like immunohistochemical, RT-PCR or Western blot are necessary in the future experiments”. Why not present the results of these experiments in this manuscript? Therefore, this manuscript data too few and less novelty.

Response:

We are aware of this limitation but carrying out experiments described at this moment is challenging as we don’t have enough resources. What is more, all reagents are not readily available in our country and would have to be imported, which would be not only costly but also time-consuming.  We however appreciate your suggestions and will try to include these procedures in the experiments we carry out in the future.

Reviewer 2 Report

The present manuscript present results on coronary flow, nitrite and superoxide anion released in venous effluent, obtained using perfused rat heart model. Perfusion with control solution, or solutions containing sildenafil (10-200 nM) or sildenafil + L-NAME (30 µM) were tested.

The results are presented as Tables, which makes it difficult to clearly see the data set as a whole.

Several issues with the manuscript need to be addressed  :

Introduction:

The link between the first part of the introduction and the presentation of the objective of the study is not clear. Whereas the author emphasize the ischaemic/reperfusion and preconditioning strategies, the study focused only on a perfused heart model not submitted to ischaemia. It needs to be better explained, how the objective of the study aims could be valuable  for explaining the possible preconditioning properties of sildenafil.

line 38 : PAH is actually a current indication for sildenafil (supported by more than just only “emerging” evidence)

l51 : what are the reported mechanisms for “cAMP-supressing effects” of sildenafil (reference?)

Methods:

The presentation of the protocol and sequences of treatments is confusing:

                - for the control recording, has a proper vehicle solution been tested (in other words, how sildenafil was prepared/dissolved)? If no vehicle has been tested for comparison with sildenafil, please mention in the methods.

                - from line 255, the protocol must be more rigorously presented:

- “… serving for the confirmation of endothelial function integrity…” : please explain how initial autoregulation protocol gives information on the endothelial function. Autoregulation may involve both endothelial -dependent (flow-induced vasodilation) and independent (myogenic tone) mechanisms.

- lines 259 and following, there may be a mistake: sildenafil and L-NAME is duplicated in the sentence.

From this presentation, it seems that L-NAME is intercalated between 2 sildenafil concentrations and then washed out before the following sildenafil test. Then line 274, authors refers to a “second series of experiments”, with L-NAME added at the start of the experiment. Please clarify.

- Please mention if only one, or several sildenafil concentrations were tested for each heart preparation.

As sildenafil recording were performed in series following the control recording (ending at 120 mmHg”), was it verified that the flow recovered the normal “40 mmHg” value before testing sildenafil?

When adding sildenafil, was the change in flow transient or stabilized after few seconds/minutes? Vasodilatory response may be followed by autoregulation which may have offset the CF change. The way time points (maximum or steady-state CF value) were chosen must be explained in the methods.

Results:

It would be nice to see the data as graphs, showing CF vs CPP for each pharmacological condition. This may display see the autoregulation occurring at 60 and 80 mmHg more easlily (CF seems to be stable stable at these 2 CPPs).

The methods mention ANOVA analysis but it is not clear in the Table legends when this was used.

I wonder if there is not a multiple comparison issue (alpha risk inflation) in Tables 3, 4, 5, where even control values for one given pressure are very variable. Discrete significant differences (e.g. Table 4, 20nM sildenafil+LNAME vs Control)  may appear just from random variations, given that many t-test were probably performed (alpha risk inflation). Global, 2-way analysis of sildenafil effect or pressure effect, followed by appropriated post-hoc analysis, may be necessary.

When no significant difference is found, please do not state that the values are “increased” or “decreased”.

Discussion:

The reason why differences in the recordings are observed between 10 and 20 nM, which are very close concentration, must be discussed. Please recall the expected IC50 of sildenafil on PDE5.

Discuss more rigourously how “myocardial cGMP” could influence CF values at various CPPs (mechanisms, bibliographic references?).

Sildenafil may act on cardiac, smooth muscle or endothelial cells. Please take this into account to discuss the results more rigorously. This may help to interpret the effects of L-NAME.

How does sildenafil and L-NAME treatment influence heart rate and inotropism?

L-NAME results may indicate some NOS-dependent, and Nos-independent mechanisms. Does 30 µM LNAME sufficient to inhibit all 3 isoforms of NOS?  L-NAME is described to be more potent for eNOS and nNOs compared to iNOS (Furfine, E. S., Harmon, M. F., Paith, J. E. & Garvey, E. P. Selective inhibition of

constitutive nitric oxide synthase by l-NG-nitroarginine. Biochemistry 32,8512–8517 (1993)).

l168 : how changes in gene expression may explain the effect of sildenafil observed here : time course of these effect is probably not consistent with change in gene expression. “explanation” (l171) is not convincing.

How sildenafil could increase nitrite release? Direct or indirect mechanism? It seems that main factor that increase nitrite release is CPP (or flow); please analyse these factors using statistical tests.

End of discussion in particular is full of typos, and some sentences are nonsense and need revision. Avoid non-formal forms (it’s..).

How one may interpret the consequences observed at various pressure, regarding physiological context and/or pathological? Which CPP is more physiologically relevant?

Author Response

We would first like to thank you for the time and effort it took to review our manuscript. We appreciate all of your comments and constructive criticism. Please find point-by-point all the changes we have made to the manuscript, according to your suggestions.

Comment 1:

The link between the first part of the introduction and the presentation of the objective of the study is not clear. Whereas the author emphasizes the ischemic/reperfusion and preconditioning strategies, the study focused only on a perfused heart model not submitted to ischemia. It needs to be better explained, how the objective of the study aims could be valuable for explaining the possible preconditioning properties of sildenafil.

Response:

The aim of our study was to evaluate effects of different doses of sildenafil on coronary flow, nitrite levels and oxidative stress markers, as we assessed from the literature that those were the main cardioprotective properties of sildenafil and other PDE5 inhibitors, that could have role in the pharmacological preconditioning. Our goal was to analyze concentrations of sildenafil at which those properties were the most pronounced, which could help us to narrow the range of sildenafil concentrations with the most potent effect on preconditioning.

Comment 2:

Line 38: PAH is actually a current indication for sildenafil (supported by more than just only “emerging” evidence).

Response:

We researched the literature and corrected that sentence accordingly.

Comment 3:

L51: what are the reported mechanisms for “cAMP-supressing effects” of sildenafil (reference?)

Response:

We added references to noted statement.      

Comment 4:

For the control recording, has a proper vehicle solution been tested (in other words, how sildenafil was prepared/dissolved)? If no vehicle has been tested for comparison with sildenafil, please mention in the methods.

Response:

Sildenafil solution was made by dissolving sildenafil in dimethyl sulfoxide (DMSO) and diluting with perfusion buffer to the final concentration of DMSO of 0.001%.  For control buffer solution, the same volume of DMSO was added.

Comment 5:

…”serving for the confirmation of endothelial function integrity…”: please explain how initial autoregulation protocol gives information on the endothelial function. Autoregulation may involve both endothelial- dependent (flow-induced vasodilation) and independent (myogenic tone) mechanisms.

Response:

We have reformulated our statement according to your observation, and after consulting literature.

Comment 6:

Lines 259 and following, there may be a mistake: sildenafil and L-NAME is duplicated in the sentence.

Response:

We deleted the duplicated “sildenafil and L-NAME” from those lines.

Comment 7:

From this presentation, it seems that L-NAME is intercalated between 2 sildenafil concentrations and then washed out before the following sildenafil test. Then line 274, authors refer to a “second series of experiments” with L-NAME added at the start of the experiment. Please clarify.

Response:

Thank you for noticing this confusing presentation of our methodology. One rat heart was used only for one experimental protocol- ether for sildenafil or sildenafil with L-NAME. Testing combination of sildenafil and L-NAME were in fact the second series of experiments, where after the heart stabilization of 30 minutes, L-NAME was administered (30 μM). We waited for 5 minutes so there would be enough time for inhibition of NO synthase. After 5 minutes, sildenafil was added, waited for another 5 minutes and then parameters of myocardial function at different CPP were measured (Figure 7).

Comment 8:

Please mention if only one, or several sildenafil concentrations were tested for each heart preparation.

Response:

We tested every rat heart with only one sildenafil dose, or one sildenafil dose + L-NAME. We stated that now in the Materials and Methods section, as you suggested.

Comment 9:

As sildenafil recording were performed in series fallowing the control recording (ending at 120 mm Hg), was it verified that the flow recovered the normal “40 mm Hg” value before testing sildenafil?

Response:

Before administration of any medication, we took measurements at baseline at different pressures (40, 60, 80, 120 mm Hg). After the highest pressure of control-testing, pressure was recovered to the lowest (40 mm Hg), from which we started the administration of sildenafil at different doses. We explained that in detail in our Materials and Methods section, as you suggested.

Comment 10:

When adding sildenafil, was the change in flow transient or stabilized after few seconds/minutes? Vasodilatory response may be followed by autoregulation which may have offset the CF change. The way time points (maximum or steady-state CF value) were chosen must be explained in the methods.

Response:

After the control testing, pressure was recovered to the basepoint (40 mm Hg). Then, five minutes were waited before administration of sildenafil or L-NAME. We waited again for five minutes, so that we can get steady CF value that would be recorded at the end of those five minutes. CPP would then be raised for 20 mm Hg, when we repeated the whole process. After reaching CPP of 120 mm Hg, and  the results recording, the experiment was finished for that one isolated rat heart. We explained this whole process in the Material and Methods section, and presented it via Graphic 7 for better understanding of the experimental protocol.

Comment 11:

It would be nice to see the data as graphs, showing CF vs CPP for each pharmacological condition. This may display see the autoregulation occurring at 60 and 80 mmHg more easily (CF seems to be stable at these 2 CPPs).

Response:

The results have now been presented in bar graphs, including symbols for statistical analysis. We believe that the graphical presentation of findings now makes it easier to observe the actual findings, and we thank you again for your suggestion on how to improve our presentation of results.

Comments 12 and 13:

The methods mention ANOVA analysis, but it is not clear in the Table legends when this was used.

I wonder if there is not a multiple comparison issue (alpha risk inflation) in Tables 3, 4, 5, where even control values for one given pressure are very variable. Discrete significant differences (e.g. Table 4, 20nM sildenafil + L-NAME vs Control) may appear just from random variations, given that many t-test were probably performed (alpha risk inflation). Global, 2-way analysis of sildenafil effect or pressure effect, followed by appropriated post hoc analysis, may be necessary.

Response:

As it was stated in statistical analysis part of Material and Methods section, we did a mixed between-within subject ANOVA. In the other words, we did comparison between every CPP value (40, 60, 80, 100, 120 mm Hg) as well as between control and experimental group. Statistical significances noted in graphics represent the post-hoc analysis (Tukey’s HSD) between control and experimental group, once the ANOVA was confirmed as statistically significant.

Comment 14:

When no significant difference is found, please do not state that the values are “increased” or “decreased”.

Response:

We reformulated or deleted sentences that were misleading, not commenting on an increase or a decrease where statistical significance was found.

Comment 15:

The reason why differences in the recording are observed between 10 and 20 nM, which are very close concentration, must be discussed. Please recall the expected IC50 of sildenafil on PDE5.

Response:

A review of the literature showed that the measured values for the potency and selectivity of PDE-5 inhibitors vary, with the IC50 values on PDE5 for sildenafil ranging from 3.5 nM to 8.5 nM (18). We have selected our concentration ranges based on previous studies and having in mind the IC50 values from the literature (11), with the minimum tested concentration expected to induce PDE5 inhibition, and provide measurable effect of tested parameters in our study design. Regarding the obtained differences in results of our study for two close concentrations, we must again make clear that each heart was tested only for a single dose of sildenafil, so for these two close concentrations, differences might have been the consequence of different basal levels of cGMP or activity of guanylyl cyclase and other steps in the signaling cascade. However, with higher tested concentrations (50-200nM), differences between control and experimental conditions more consistent. In interpretation of findings, we have now included information that observed effects suggest that effects of sildenafil may involve not only PDE5 inhibition, but also the inhibition of other PDE types. It was already reported in the literature that the increase in sildenafil dose and concentration decreases its specificity for only PDE5 inhibition, and other PDE subtypes could be inhibited(18). The highest concentration of sildenafil that we used in our experiment (200nM) showed negative effects on coronary flow, which is similar with results of the other studies, where 50nM concentration improved reperfusion function and reduced infarct size, while the concentration of 200nM led to exacerbation of ischaemic/reperfusion injury (11).

Comment 16:

Discus more rigourously how “myocardial cGMP” could influence CF values at various CPPs (mechanisms, bibliographic references?).

Response:

NO is the molecule that can trigger the cardioprotective cGMP-PKG pathway. NO first activates soluble guanylate cyclase (sGC), which catalyses synthesis of cGMP. Elevated intracellular level of cGMP mostly targets and activates cGMP–dependent protein kinase (PKG). PKG then lower the blood pressure through vasodilatation (21). As the ischaemia effects on the content of myocardial cGMP, it is shown in the isolated heart model, that cardiomyocytes have the ability of increasing cGMP levels during the first 10-25 minutes of ischaemia, with acute decrease after that period (22). Although number of studies tried to stimulate the cGMP synthesis, preservation of cGMP by blockade of PDE5 demonstrated to limit the infarct size in different studies (23). Even though results from the study conducted by Elrod et. al (24) showed no elevation of cGMP levels after sildenafil administration, they hypothesize that subcellular localization of various cGMP pools (soluble cGMP and particulate cGMP) could have a significant place in cGMP/PKG pathway of cardioprotection (24).

Comments 17, 19 and 21:

Sildenafil may act on cardiac, smooth muscle or endothelial cells. Please take this into account to discuss the results more rigorously. This may help to interpret the effects of L-NAME.

L-NAME results may indicate some NOS-dependent, and NOS-independent mechanisms. Does 30 µM L-NAME sufficient to inhibit all 3 isoforms of NOS? L-NAME is describet to be more potent for eNOS and nNOs compared to iNOS (Furfine, E.S., Harmon, M.F., Paith, J.E. & Garvey, E.P. Selective inhibition of constitutive nitric oxide synthase by l-NG-nitroarginine. Biochemistry 32, 8512-8517 (1993)).

How sildenafil could increase nitrite release? Direct or indirect mechanism? It seems that main factor that increase nitrite release is CPP (or flow); please analyze these factors using statistical tests.

Response:

Nitrite increment retained in group treated with 20 nM sildenafil despite the inhibition of the NOS system by L-NAME. This can be the consequence of the sildenafil-mediated cardioprotection independent of the iNOS, eNOS and the cGMP pathway, as tested in the study from United States (24). In that study, they used iNOS and eNOS knockout mice in which they induced an infarct, and threated them with sildenafil afterwards (24). Their results showed reduction in infarct per area at risk (Inf/AAR) in mice, despite the lack of iNOS or eNOS (24), suggesting that sildenafil cardioprotective mechanisms are not only iNOS and eNOS related. Thus, the inhibition of nitric oxide synthase system by L-NAME was overcomed by those other mechanisms of action of sildenafil, which resulted in increase of NO concentration in experimental group, compared to the control group. Considering that, the reason why sildenafil significant effects on the NO concentration was only observed in the 20 nM group, and not in the others, could be explained by L-NAME wide, nonspecific and still unknown effects beside the nonspecific inhibition of NOS (25). Potential direct effects on cardiac tissue as hypothesized by some authors (25), could be one of those properties of L-NAME that interfere with cardioprotective effects of sildenafil. 

Comment 18:

How does sildenafil and L-NAME influence heart rate and inotropism?

Response:

As the effects on heart rate and inotropism of sildenafil and L-NAME were not investigated in our study, we have not mentioned this in the discussion, but considering the importance of this, we have now included a short paragraph touching on these effects.

“Previous studies have reported on the rate reduction effect of sildenafil, partially attributed to the NO synthase-mediated signaling pathways, showed to play an important role in pulmonary vein arrhythmogenesis (36). Sildenafil was also shown to have a negative chronotropic effect on mice sino-atrial nodes (37). This implies that sildenafil may potentially protect from atrial fibrillation. On the contrary, some literature suggests that sildenafil alone as well as the combination of sildenafil and L-NAME, does not affect heart rate in rats (38,39). Furthermore, higher concentrations of sildenafil (0.2 µg/ml), with a nitric oxide (NO) donor, increased ventricular tachycardia or fibrillation, implying a potential risk of atrial arrhythmogenesis (40). Also, studies conducted on dogs and humans found that sildenafil does not affect cardiac contractility (inotropism) (41), despite the fact that in addition to direct effects on cardiomyocytes, inhibitors of different PDE subtypes regulate vascular activity, as well as cardiac contractility. However, sildenafil was found to increase the pulmonary vein diastolic tension, but decrease contractility, which was attributed to the electrophysiological effects of sildenafil via mechanoelectrical feedback (36). Also, PDE5A inhibition by sildenafil was found to blunt out systolic responses to beta-adrenergic stimulation in a clinical study, supporting the role of the PDE5A activity in modifying stimulated cardiac function (42).”

Comment 20:

L168: how changes in gene expression may explain the effect of sildenafil observed here: time course of these effect is probably not consistent with change in gene expression. “explanation” (l171) is not convincing.

Response:

We deleted that part of our discussion, as it would bring more questions than answers regarding the sildenafil effects and gene expression.

Comment 22:

End of discussion in particular is full of typos, and some sentences are nonsense and need revision. Avoid non-formal forms (it’s…).

Response:

We thoroughly read the end as well as the other parts of Discussion and the manuscript, corrected typos and rephrased sentences that were confusing. 

Comment 23:

How one may interpret the consequences observed at various pressure, regarding physiological context and/or pathological? Which CPP is more physiologically relevant ?

Response:

Taking into account that coronary vasculature fills during the diastole (80 mm Hg) (45), results observed at CPP 60-100 mm Hg should have the greatest implications for future clinical studies, with 10 nM and 20 nM concentrations that could be used as a starting dosage of sildenafil, considering that the most of the statistically significant results from our study were observed with that doses, at 60 – 100 mm Hg of CPP. 

Reviewer 3 Report

The authors of the manuscript entitled “The effects of different doses of sildenafil on coronary flow and oxidative stress in isolated rat heart“ evaluated the dose-response relationship of sildenafil effects on cardiac function. Very interesting and informative results of the study implied that Sildenafil administration resulted in increased coronary flow at lower perfusion pressure values at all administered doses, while with higher values of coronary perfusion pressure, a reduction in coronary flow was observed, particularly expresses as NO increase in coronary venous effluent. Moreover, after the inhibition of the NOS system by L-NAME, only a dose of 200 nM sildenafil was high enough to overcome the inhibition and boost the release of superoxide. The study results proved the dose-depended effects of sildenafil and the spectrum of coronary vasculature reactions from vasodilatation at lower perfusion pressures to vasoconstriction with pressure increase.

The focus of the present investigation deserves special attention regarding the usage of PDE5 inhibitors and cardiac effects in the human population, and provides a detailed analysis of oxidative stress markers in the coronary circulation, as well as the coronary flow, after administration of sildenafil and inhibitor of NOS system – L-NAME. This manuscript explored in detail the specific connection between two pharmacological agents on retrogradely perfused isolated rat hearts allowing the readers to have a deeper insight into this actual, but worrying issue such as sildenafil effect on the heart.

The article should be accepted for publication in its current form.

Author Response

We would first like to thank you for the time and effort it took to review our manuscript. We appreciate all of your comments and constructive criticism. Thank you also for suggesting our paperwork for publication without any further corrections.

Sincerely,

The authors. 

Reviewer 4 Report

The study by Banjac et al. titled “THE EFFECTS OF DIFFERENT DOSES OF SILDENAFIL ON CORONARY BLOOD FLOW AND OXIDATIVE STRESS IN ISOLATED RAT HEART” aimed to examine the dose-dependent effect of sildenafil on coronary flow and oxidative stress markers in isolate rat heart.  The work seems to be carefully performed, however, I have few comments which should be considered by the authors.

Comments:

1.      Please be consistent with the using of abbreviations, they should be defined when they arrive first time in the text, in some cases the explanations are missing. Lines: 19, 35, 45, 48, 82 etc.

2.      Generally, in the Tables the meaning of x±SD is not explained. It supposed to be mean±SD?

3.      In lines 106-108, the interpretation of the results is not appropriate. Sildenafil did not increase the nitrite levels significantly (at dose of 20nM in 120mmHg was even a decrease of them after sildenafil), thus the author cannot declare that the sildenafil administration led to rise” of nitrite level. It should be reformulated. If there was a trend to increase the nitrite level, is should be proven by addition of the exact value of P.

4.      The same is valid for the Table 4 and 5. The data are heterogenous, they cannot be interpreted in the presented fashion. Lines 111-114: the declaration “sildenafil and L-NAME combination released smaller amounts of nitrite” has to be proven statistically.  The data of the Table 4 should be more accurately described as well (Lines: 128-130).    

5.      In lines 173-175 is mentioned that the cardioprotective effect of sildenafil in not only NO-dependent. It would be interesting to add other possible signaling pathways or mechanisms which could be involved.

6.      The flowmetric method should be described in more detail, the used devises (probes, transducers) should be added.

Author Response

We would first like to thank you for the time and effort it took to review our manuscript. We appreciate all of your comments and constructive criticism. Please find point-by-point all the changes we have made to the manuscript.

Comment 1:

Please be consistent with the using of abbreviations, they should be defined when they arrive first time in the text, in some cases the explanations are missing. Lines: 19, 35, 45, 48, 82 etc.

Response:

All abbreviations have been defined at their first appearance in the text, as suggested.

Comment 2:

Generally,  in the Tables the meaning of x±SD is not explained. It supposed to be mean±SD?

Response:

We changed every “x ± SD” to “mean ± SD” wherever it appeared in our manuscript, so it won’t be confusing.

Comment 3:

In lines 106-108, the interpretation of the results is not appropriate. Sildenafil did not increase the nitrite levels significantly (at dose of 20 nM in 120 mmHg was even a decrease of them after sildenafil), thus the author cannot declare that the sildenafil administration “led to rise” of nitrite level. It should be reformulated. If there was a trend to increase the nitrite level, is should be proven by addition of the exact value of P.

Response:

We deleted that disputed sentence since we didn’t find statistically significant difference in nitric-oxide levels between control group and the sildenafil-threated groups. Therefore we cannot say that sildenafil led to an increase in NO concentrations, as you observed.

Comment 4:

The same is valid for the Table 4 and 5. The data are heterogenous, they cannot be interpreted in the presented fashion. Lines 111-114: the declaration “sildenafil and L-NAME combination released smaller amounts of nitrite” has to be proven statistically. The data of the Table 4 should be more accurately described as well (Lines: 128-130).

Response:

We reformulated or deleted sentences that were misleading if the obtained results of our experiment were the effects of sildenafil (or sildenafil + L-NAME) or not, according the statistical significance of those results.

Comment 5:

In lines 173-175 is mentioned that the cardioprotective effect of sildenafil in not only NO-dependent. It would be interesting to add other possible signaling pathways or mechanisms which could be involved.

Response:

Nitrite increment retained in group treated with 20 nM sildenafil despite the inhibition of the NOS system by L-NAME. This can be the consequence of the sildenafil-mediated cardioprotection independent of the iNOS, eNOS and the cGMP pathway, as tested in the study from United States (24), suggesting that sildenafil cardioprotective mechanisms are not only iNOS and eNOS related. Several studies showed that opening of mitochondrial KATP channels plays an important role in pharmacological preconditioning by sildenafil. Namely, opening of the mitochondrial KATP channels can compensate the loss in membrane potential, enabling additional pumping of protons and forming and H+ gradient for ATP synthesis and Ca2+transport. The sildenafil induced protection can be induced mediated by directly opening of the mitochondrial KATP or through different signaling pathways, as protein kinase C and MAP kinases activation.

Comment 6:

The flowmetric method should be described in more detail, the used devises (probes, transducers) should be added.

Response:

We added the transducer (sensor) model in Materials and Methods section, as well as the way of its placement and the method of collecting data of myocardial function.

Round 2

Reviewer 1 Report

1. Author still don;t correct this mistakes. In the References section, the first word should be capitalized, and the other words should be lowercase, such references 3, 8, 9 etc. 

2. Please recheck the Statistical analysis at Sildenafil 10 nM group in the Figure 1. Compare the statistical difference between the two groups at CCP 40 mmHg.

Author Response

We would first like to thank you for the time and effort it took to review our manuscript. We appreciate all of your comments and constructive criticism. Please find all the changes we have made to the manuscript, according to your suggestions.

Comment 1:

Author still don;t correct this mistakes. In the References section, the first word should be capitalized, and the other words should be lowercase, such references 3, 8, 9 etc. 

Response:

We now corrected those mistakes, and now references are made according to your suggestion.

Comment 2:

Please recheck the Statistical analysis at Sildenafil 10 nM group in the Figure 1. Compare the statistical difference between the two groups at CCP 40 mmHg.

Response:

Thank you for noticing that we haven’t emphasized significance of the results of Sildenafil 10nM group at CCP 40 mm Hg, as there is statistically significant difference in the flow between control and experimental group. However, while making figures from tables, we left out that important information, by accident. We corrected that now.

Reviewer 4 Report

The authors have addressed all my comments for this paper and answered the technical questions I have for the method. The paper has been significantly improved after revising. I welcome the change of tables to figures. However have a minor comment on figures: the units on the Y axis are missing, they should be added.

Author Response

Firstly, thank you for taking time to review our manuscript. We appreciate your feed-back as well as the comments that helped us to significantly improve the clarity and quality of our paper. Please find bellow the changes we have made to the manuscript.

Comment:

The units on the Y axis are missing, they should be added.

Response:

We added units on the Y axis of our figures.